# Phase separation-mediated actin bundling by the postsynaptic density condensates

Xudong Chen[1], Bowen Jia[1], Shihan Zhu[1], Mingjie Zhang[2,3]*

[1]Division of Life Science, Hong Kong University of Science and Technology, Clear Water Bay, Kowloon, Hong Kong, China; [2]Greater Bay Biomedical Innocenter, Shenzhen Bay Laboratory, Shenzhen, China; [3]School of Life Sciences, Southern University of Science and Technology, Shenzhen, China

**Abstract** The volume and the electric strength of an excitatory synapse is near linearly correlated with the area of its postsynaptic density (PSD). Extensive research in the past has revealed that the PSD assembly directly communicates with actin cytoskeleton in the spine to coordinate activity-induced spine volume enlargement as well as long-term stable spine structure maintenance. However, the molecular mechanism underlying the communication between the PSD assembly and spine actin cytoskeleton is poorly understood. In this study, we discover that in vitro reconstituted PSD condensates can promote actin polymerization and F-actin bundling without help of any actin regulatory proteins. The Homer scaffold protein within the PSD condensates and a positively charged actin-binding surface of the Homer EVH1 domain are essential for the PSD condensate-induced actin bundle formation in vitro and for spine growth in neurons. Homer-induced actin bundling can only occur when Homer forms condensate with other PSD scaffold proteins such as Shank and SAPAP. The PSD-induced actin bundle formation is sensitively regulated by CaMKII or by the product of the immediate early gene *Homer1a*. Thus, the communication between PSD and spine cytoskeleton may be modulated by targeting the phase separation of the PSD condensates.

## Editor's evaluation

This paper used a reconstitution approach to show that in vitro reconstituted PSD condensates can promote actin polymerization and filamentous actin bundling in the absence of other actin regulatory proteins. The authors further show that the EVH1 domain of Homer is responsible for this activity, which is also regulated by CaMKII. Together, this convincing evidence provides the fundamental insight that the crosstalk between PSD and the spine cytoskeleton may be modulated by targeting the phase separation of the PSD condensates.

## Introduction

Neuronal synapses are highly dynamic in developing and in mature brains. In response to input signals, the physical sizes of a synapse can enlarge, shrink, or even disappear altogether (*Berry and Nedivi, 2017*). Such physical changes are directly correlated with strengthening, weakening, or complete loss of signal transmission of the synapse, and thus are of fundamental importance in neuronal circuit formation and brain function (*Martin et al., 2000*; *Sutton and Schuman, 2006*; *Takeuchi et al., 2014*). Taking excitatory synapses for examples, stimulation of a synapse can induce rapid (in a matter of a few minutes) but transient spine volume enlargements due to selective enrichments of proteins such as actin and its regulatory proteins accompanied by overall increase of actin cytoskeletal dynamics (*Bosch et al., 2014*; *Hering and Sheng, 2003*; *Kim et al., 2015*; *Mikhaylova et al., 2018*; *Okamoto et al., 2009*; *Okamoto et al., 2004*; *Sekino et al., 2006*). Such temporary destabilization of actin

*For correspondence: zhangmj@sustech.edu.cn

cytoskeletons is thought to be critical for subsequent overall cytoskeleton reorganization necessary for sustained spine enlargements (*Hering and Sheng, 2003*). Long-term stabilization of an enlarged spine requires synthesis of new synaptic proteins, stabilization of actin filaments, and enlargement of the postsynaptic density (PSD) (*Bramham, 2008*; *Sekino et al., 2007*). It is generally perceived that long-term information storage in a synapse not only requires a stable PSD assembly containing certain densities of clustered glutamate receptors but also needs structural support by cross-linked actin filaments (*Basu and Lamprecht, 2018*; *Dillon and Goda, 2005*; *Lamprecht, 2014*; *Lei et al., 2016*).

The PSD of a synapse is extremely dense and formed by hundreds of proteins via phase separation-mediated self-assembly beneath the synaptic plasma membranes (*Blomberg et al., 1977*; *Chen et al., 2008*; *Chen et al., 2020*; *Harris and Weinberg, 2012*; *Zeng et al., 2018*). The PSD and actin cytoskeleton are tightly associated with each other in spines. Biochemically purified PSDs are enriched with actin and actin-binding proteins (*Cheng et al., 2006*; *Sheng and Hoogenraad, 2007*). Filamentous actins (F-actin), both in linear and branched forms, build a cross-linked matrix extending from spine head toward spine cytoplasm and shape the physical structure of spines (*Burette et al., 2012*; *Cingolani and Goda, 2008*; *Fifková and Delay, 1982*). Electron microscopy (EM) studies revealed that actin cytoskeleton directly contacts and embraces PSD from the cytoplasmic side (*Korobova and Svitkina, 2010*; *Rácz and Weinberg, 2013*). Super-resolution imaging studies showed that F-actin is initially nucleated at PSD and then elongates outward to the base of dendritic spine during spine growth (*Chazeau et al., 2014*). Treatment of neurons with F-actin depolymerization drugs such as latrunculin-A disrupts the organization of PSD nanodomains formed by scaffold proteins including PSD-95, SAPAP, Shank and Homer (*Kerr and Blanpied, 2012*; *Kuriu et al., 2006*; *MacGillavry et al., 2013*). Overexpression of Homer1 in cultured neurons promotes the synaptic enrichment of F-actin (*Usui et al., 2003*). Conversely, removal or decrease of PSD scaffold proteins in neurons leads to severe synaptic growth defects, a phenotype that is directly related to dysregulation of actin cytoskeleton (*Hung et al., 2008*; *Jiang and Ehlers, 2013*; *Schmeisser et al., 2012*; *Silverman et al., 2011*; *Wöhr et al., 2011*). Autism-related mutations on Shank3 also cause altered expressions of PSD-enriched actin-binding proteins such as cofilin, cortactin, and Rac1 (*Joensuu et al., 2018*).

Although communications between PSD and actin cytoskeleton during spinogenesis have been widely observed and extensively studied, molecular mechanisms underlying such communications are poorly understood. The poor mechanistic understanding of the communication between the PSD and spine actin cytoskeleton is at least in part due to the practical challenges of the system. Both PSD and actin cytoskeleton are large and heterogeneous molecular assemblies formed by phase separation or phase transition (*Chen et al., 2020*; *Zeng et al., 2016*; *Zeng et al., 2018*). There are no suitable methods available for detailed mechanistic studies of two molecular assemblies formed by totally different modes of phase separation/transition. Another practical challenge for detailed mechanistic studies of the communication between the PSD and actin cytoskeleton is the awkward size of synapses, which is typically with a diameter below 0.5 μm and near the limit of optical microscopy resolution (*Gray, 1959*; *Palay, 1956*). It is not trivial to study the molecular mechanism governing the large PSD assembly and the surrounding F-actin network within the tiny compartment of a dendritic spine in living neurons.

Biochemically reconstituted PSDs offer a unique platform for investigating and dissecting the underlying molecular mechanisms governing the interaction between the PSD assembly and actin cytoskeleton. In this study, we unexpectedly found that the PSD condensates devoid of any known actin-binding/regulatory proteins can promote actin polymerization and F-actin bundle formation. Remarkably, the lower layer of the PSD condensates formed by SAPAP, Shank and Homer are necessary and sufficient for inducing F-actin bundle formation, explaining that F-actin nucleates at the cytoplasmic side of PSD and emanates toward spine cytoplasm during synaptic growth. We further discovered that Homer and its EVH1 domain are essential for the PSD condensate-induced F-actin bundling. However, Homer can induce actin bundling only when it forms phase separated condensates with other PSD scaffold proteins. Accordingly, the communication between the PSD and actin cytoskeleton can be modulated by regulating the PSD condensate formation or dispersion.

## Results

### PSD condensate-induced F-actin bundle formation

Our previous study showed that the PSD condensates formed with multiple postsynaptic scaffold proteins (including PSD-95, SAPAP, Shank and Homer) could robustly induce bundling of F-actin in an in vitro reconstruction assay (*Zeng et al., 2018*). Unexpectedly, such PSD condensate-mediated F-actin bundling was independent of actin regulatory proteins, as removal of the Arp2/3 complex from PSD condensates hardly affected F-actin bundling. This finding suggests that PSD condensates may directly induce F-actin bundling.

We took advantage of the biochemically reconstituted PSD system with defined components (*Zeng et al., 2018*; *Zeng et al., 2019*) to test the above hypothesis. We first assembled PSD condensates composed of six major synaptic proteins (Stg, PSD-95, SynGAP, SAPAP1 [also known as GKAP], Shank3 and Homer3, each at 10 µM; the system is termed as 6× PSD), and then mixed the pre-formed PSD condensates with 5 µM G-actin (*Figure 1A*). Differential interference contrast (DIC) and fluorescent microscopic imaging showed that rhodamine-labeled G-actin was rapidly enriched into PSD condensates (marked with iFluor-488-labeled PSD-95) upon mixing. Interestingly, dense actin signals continue to grow into fibrous structures after the mixing, but the PSD-95 signal remains as droplets (*Figure 1B*). This observation is reminiscent of an earlier super-resolution imaging-based finding showing that, in cultured neurons, synaptic actin was first concentrated and nucleated at PSD and F-actin then emanated from PSD toward spine cytoplasm (*Chazeau et al., 2014*). It is noted that, in our reconstitution assay, actin was polymerized and bundled in the absence of any known actin regulatory molecules, suggesting that the PSD condensates can directly induce actin bundle formation.

We next compared the distribution pattern of each PSD component with that of actin bundles in the actin/PSD condensate mixtures. To avoid potential cross-talks between fluorophores, we separated the assay into three parallel groups with only three proteins labeled with different fluorophores in each group (two synaptic proteins labeled with Alexa-647 and iFluor-488 plus rhodamine-labeled actin) (*Figure 1C–E*). Imaging-based assay revealed two distinct distribution patterns for the 6× PSD proteins: a droplet-like pattern for Stg, PSD-95, and SynGAP and these proteins only partially co-localize with the fibrous F-actin bundles; a fibrous pattern for GKAP, Shank3, and Homer3 and these proteins largely overlap with the F-actin bundles (*Figure 1C–E*). Our earlier study showed that Stg, PSD-95, and SynGAP form the upper layer (i.e., the layer closer to the PSD membrane) of the PSD assembly and GKAP, Shank3, and Homer3 form the lower layer of the PSD with the two layers connected by GKAP (*Zeng et al., 2018*). The imaging data shown in *Figure 1C–E* seem to imply that the lower layer of the PSD, instead of the upper layer, directly interacts with and modulates the formation of actin bundles (also see *Figure 1H*). Interestingly, it has been shown that actin filaments in dendritic spine do not directly contact with proteins proximal to postsynaptic membranes (e.g., Stg and PSD-95). Instead, actin filaments interface with synaptic proteins such as Shank and Homer that reside at the deeper layer of the PSD (*Hotulainen and Hoogenraad, 2010*; *Westin et al., 2014*).

### F-actin bundling requires the phase separation of Homer

Consistent with the data in *Figure 1C–E*, a Stg, PSD-95, and SynGAP mixture (each at 10 µM) underwent phase separation but could not induce actin bundle formation (*Figure 1F1*, quantified in *Figure 1G*). In contrast, a GKAP, Shank3, and Homer3 mixture (each at 10 µM) strongly induced F-actin bundling (*Figure 1F2* and *Figure 1G*), indicating that the lower layer of PSD proteins interacts with actin. Dropping Homer3 out from the GKAP, Shank3, and Homer3 mixture eliminated the actin bundle inducing capacity of the mixture, although the binary GKAP and Shank3 solution was still capable of undergoing phase separation (*Figure 1F3* and *Figure 1G*). Thus, Homer3 is an essential element for the low layer PSD condensates to induce actin bundling. We further showed that Homer1 could replace Homer3 for the GKAP, Shank3, and Homer3 mixture to induce actin bundle formation (*Figure 1F4* and *Figure 1G*), indicating that all members of the Homer family proteins, when together with Shank and GKAP, can induce actin bundle formation. As a control, individual GKAP, Shank3, or Homer1 (each at 10 µM) could not induce actin bundle formation (*Figure 1—figure supplement 1A*). To check whether higher concentrations of total proteins may also facilitate F-actin bundling, we mixed 5 µM actin with 60 µM Homer3 or 60 µM BSA. No F-actin bundling was observed in either condition (*Figure 1—figure supplement 1B*). However, F-actin bundling still occurred when 3× PSD (GKAP, Shank3, and Homer1) were decreased from 10 µM to 2.5 µM (*Figure 1—figure supplement 1C*).

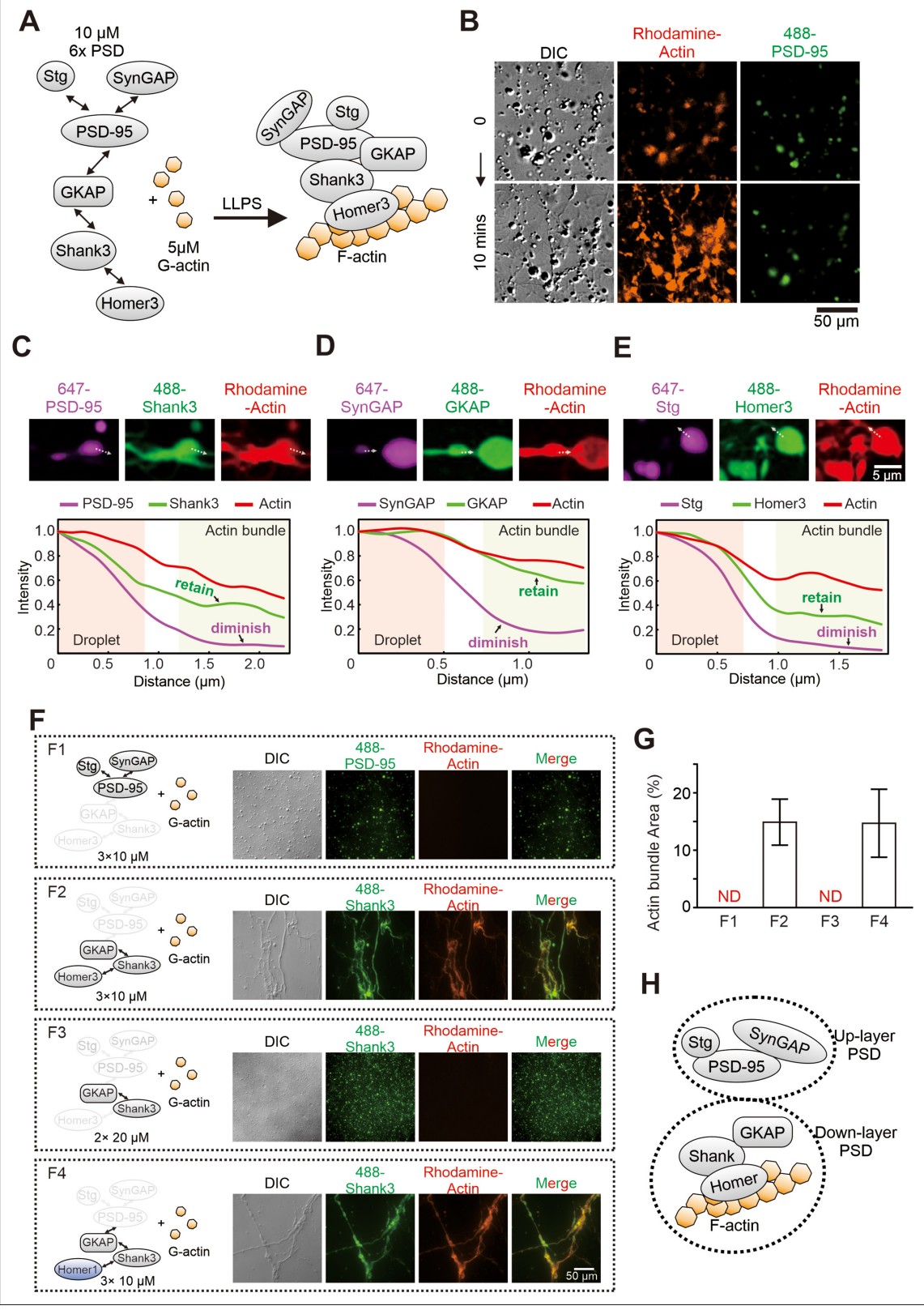

**Figure 1.** Postsynaptic density (PSD) condensate-mediated actin bundling requires Homer. (**A**) Schematic diagram showing the molecular components of the PSD protein network for the actin bundling assay. The interaction details of reconstituted PSD are indicated by two arrow-headed lines. All PSD proteins were at 10 µM and were mixed with 5 µM G-actin (all in final concentrations in the assay mixtures throughout this work). (**B**) Time-lapse differential interference contrast (DIC)/fluorescent microscopy images showing that PSD condensates (indicated by iFluor-488-PSD-95, 2% labeled)

*Figure 1 continued on next page*

*Figure 1 continued*

enriched actin (rhodamine-labeled, 5% labeled) and promoted actin polymerization. (**C–E**) Confocal images and fluorescence intensity line-plots showing that actin bundles co-enriched with Shank3, GKAP, and Homer3, but not with PSD-95, SynGAP, or Stg. All visualized PSD proteins were labeled at 2% with the indicated fluorophores. This protein labeling ratio was used throughout the study unless otherwise stated. (F and G) Representative DIC/ fluorescent microscopy images and quantification data showing the actin budling capacities by four combinations of different PSD components. F1: Stg+PSD-95+SynGAP, each at 10 µM; F2: GKAP+Shank3+Homer3, each at 10 µM; F3: GKAP+Shank3, each at 20 µM; F4: GKAP+Shank3+Homer1, each at 10 µM. The concentration of actin was kept at 5 µM in all four conditions. N=5 independent batches of imaging assays and represented as mean ± SD. ND stands for not detectable. (**H**) A model depicting the assembly of PSD proteins into two layers of condensates and interaction of actins with the lower layer of PSD condensates.

The online version of this article includes the following figure supplement(s) for figure 1:

**Figure supplement 1.** 3× Postsynaptic density (PSD) condensates, instead of individual component, bundle actin.

**Figure supplement 2.** CaMKIIβ can bundle actin filaments but with much weaker capacity than the postsynaptic density (PSD) condensates.

Taken together, the above imaging-based assays revealed that the Homer family scaffold proteins, upon forming condensates with other PSD proteins via phase separation, can induce actin bundle formation. The F-actin bundling activity of the Homer-containing PSD condensates is stronger than that of CaMKIIβ, which is a well-known F-actin-binding/bundling protein in synapses (*Shen et al., 1998*; *Figure 1—figure supplement 2*).

## Homer uses the EVH1 domain to bundle actin upon phase separation

We next searched for the molecular mechanisms underlying Homer-dependent actin bundling by the PSD condensates. Homer is a multi-domain scaffold protein in PSD. All three Homer family members (Homer1, -2 and -3) share a similar domain organization, each with an N-terminal EVH1 (Ena/Vasp homology domain 1) domain, an unstructured linker, and a C-terminal coiled-coil (CC) domain (*Figure 2A*). The EVH1 domain of Homer binds to the proline-rich motifs (PRMs) of different synaptic targets (e.g., Shank, mGluR, IP$_3$R, etc.) (*Peterson and Volkman, 2009*; *Shiraishi-Yamaguchi and Furuichi, 2007*), and the C-terminal CC is responsible for forming Homer tetramer (*Hayashi et al., 2009*). To locate the region in Homer responsible for actin bundling, we designed two Homer3 chimeras, each with a comparable phase separation capacity with that of the WT protein (*Figure 2B*). Such design allowed us to uncouple possible direct Homer/actin interaction with the phase separation of the system. For the first chimera, we replaced Homer3 CC with the CC domain of the tetrameric GCN4 (*O'Shea et al., 1991*) to probe a possible role of the Homer3 CC domain in actin bundling (*Figure 2B*, middle panel). The second chimera was designed to probe a possible role of the EVH1 domain of Homer3 in actin bundling. We substituted the EVH1 domain with the second SH3 domain from RIM-binding protein (named as rSH3), which is a presynaptic active zone protein. Concomitantly, we replaced the Homer-binding sequence (HBS) of Shank3 with the corresponding rSH3 binding PRM from RIM (*Figure 3B*, right panel). Since rSH3 binds to PRM of RIM with an affinity very similar to that of Homer3 EVH1 to HBS (*Wu et al., 2019*), the mode of the interaction between the Homer3 chimera to the Shank3 mutant should be very similar to that between the WT Homer3 and Shank3. As expected, the designed Homer3 chimeras each formed condensates with Shank3 and GKAP as WT Homer3 did (*Figure 2B*). However, only the EVH1-GCN4 chimera of Homer3, when together with Shank3 and GKAP, was capable of strongly inducing actin bundle formation. The rSH3-CC chimera of Homer3 totally lost its capacity in inducing actin bundle formation (*Figure 2C and D*). Thus, we conclude that the EVH1 domain of Homer is essential for the PSD condensate-induced F-actin bundling.

## The R3E mutation selectively impairs the actin bundling capacity of Homer1

Analysis of the Homer EVH1 domain structure revealed that it contains three highly conserved Arg residues (Arg42, Arg46, and Arg81 of Homer1, denoted as '3× Arg cluster') clustered on the surface opposite to the PRM-binding pocket of the domain (*Figure 2E*, *Figure 2—figure supplement 1*). We hypothesized that the Arg-containing charged surface of EVH1 might be involved in the Homer-actin interaction. To test this hypothesis, we performed phase separation experiments to assay actin bindings of WT Homer1 or a Homer1 mutant with the three Arg residues replaced with Glu ('Homer1-R3E'). Consistent with the structural analysis of the Homer EVH1 domain (*Figure 2E*), isothermal titration calorimetry (ITC)-based binding experiments revealed that the R3E mutation did not alter the binding

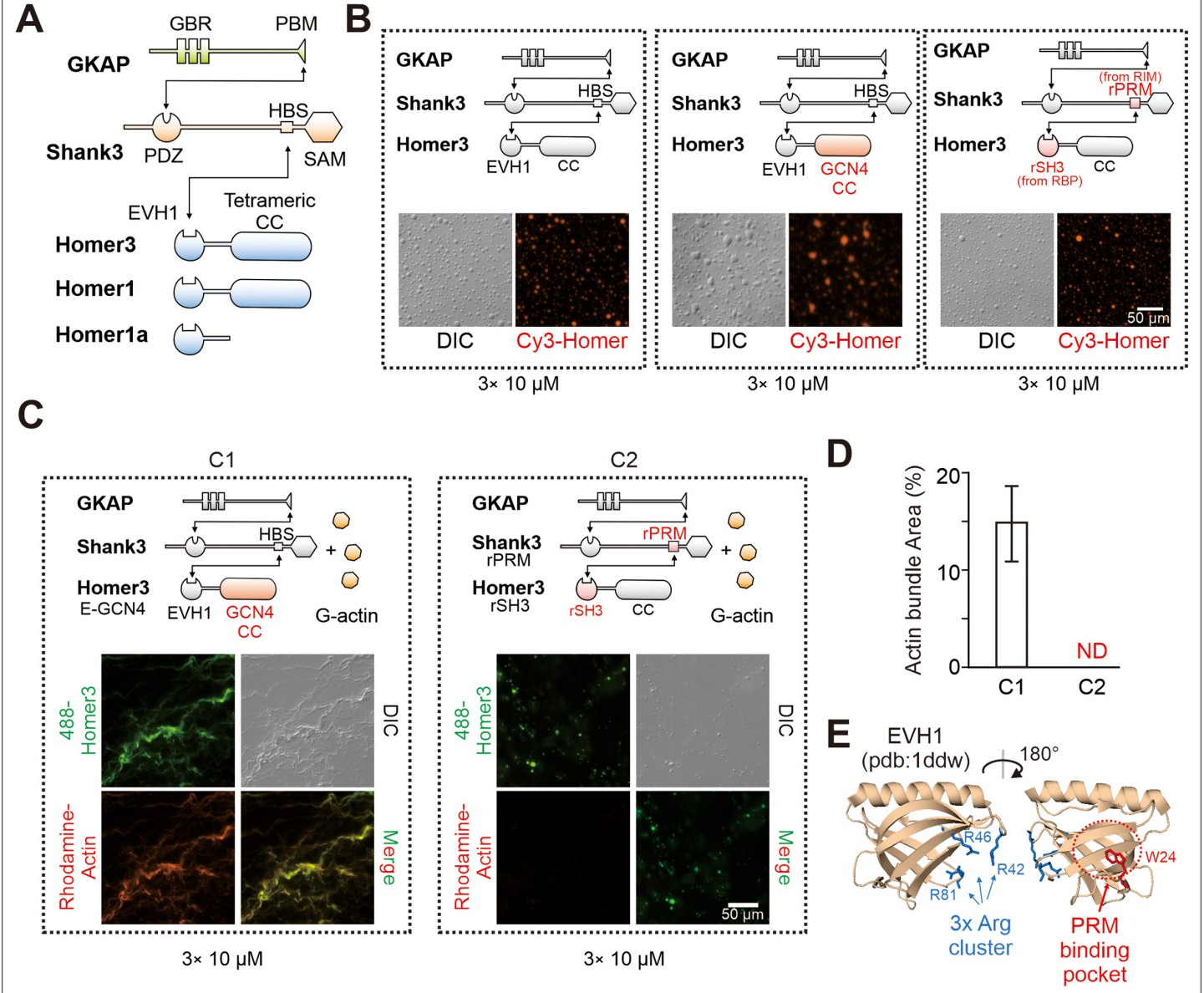

**Figure 2.** Homer in postsynaptic density (PSD) condensates uses its EVH1 domain to bundle actin. (**A**) Schematic diagram showing the domain organizations and detailed interactions among GKAP, Shank3, and Homer3. The black lines with arrows connecting protein domains/motifs depict their direct interactions. (**B**) Schematic diagrams and representative images showing that the two designed Homer3 chimeras, EVH1-GCN4 (middle) and rSH3-CC (right), could undergo phase separation with GKAP and Shank as Homer3 WT (left) did. The concentrations of all protein were fix at 10 μM. (**C** and **D**) Representative differential interference contrast (DIC)/fluorescent microscopy images and quantitative data showing that Homer3 EVH1-GCN4, but not Homer3 rSH3-CC, promoted actin bundling when mixing with GKAP and Shank3. The concentrations of all PSD proteins were fixed at 10 μM, and G-actin was used in 5 μM. N=5 independent batches of imaging assays and are presented as mean ± SD. (**E**) Structure analysis of EVH1 from Homer1 indicating a conserved (also see in *Figure 2—figure supplement 1*) positive charged surface away from the proline-rich motif (PRM)-binding pocket.

The online version of this article includes the following figure supplement(s) for figure 2:

**Figure supplement 1.** Conservation and structural analysis of the Homer EVH1 domains and representative EVH1 domains from other proteins.

of Homer1 to Shank3 (*Figure 3A*). Additionally, both imaging- and sedimentation-based phase separation assays revealed that Homer1-WT and Homer1-R3E had similar phase separation capability with Shank3 and GKAP (*Figure 3B,C*), indicating that the R3E mutation does not impact the phase separation properties of Homer1. Satisfyingly, when mixed with Shank3 and GKAP, Homer1-R3E displayed a dramatically reduced actin bundling activity compared to Homer1-WT (*Figure 3D and E*). Similarly,

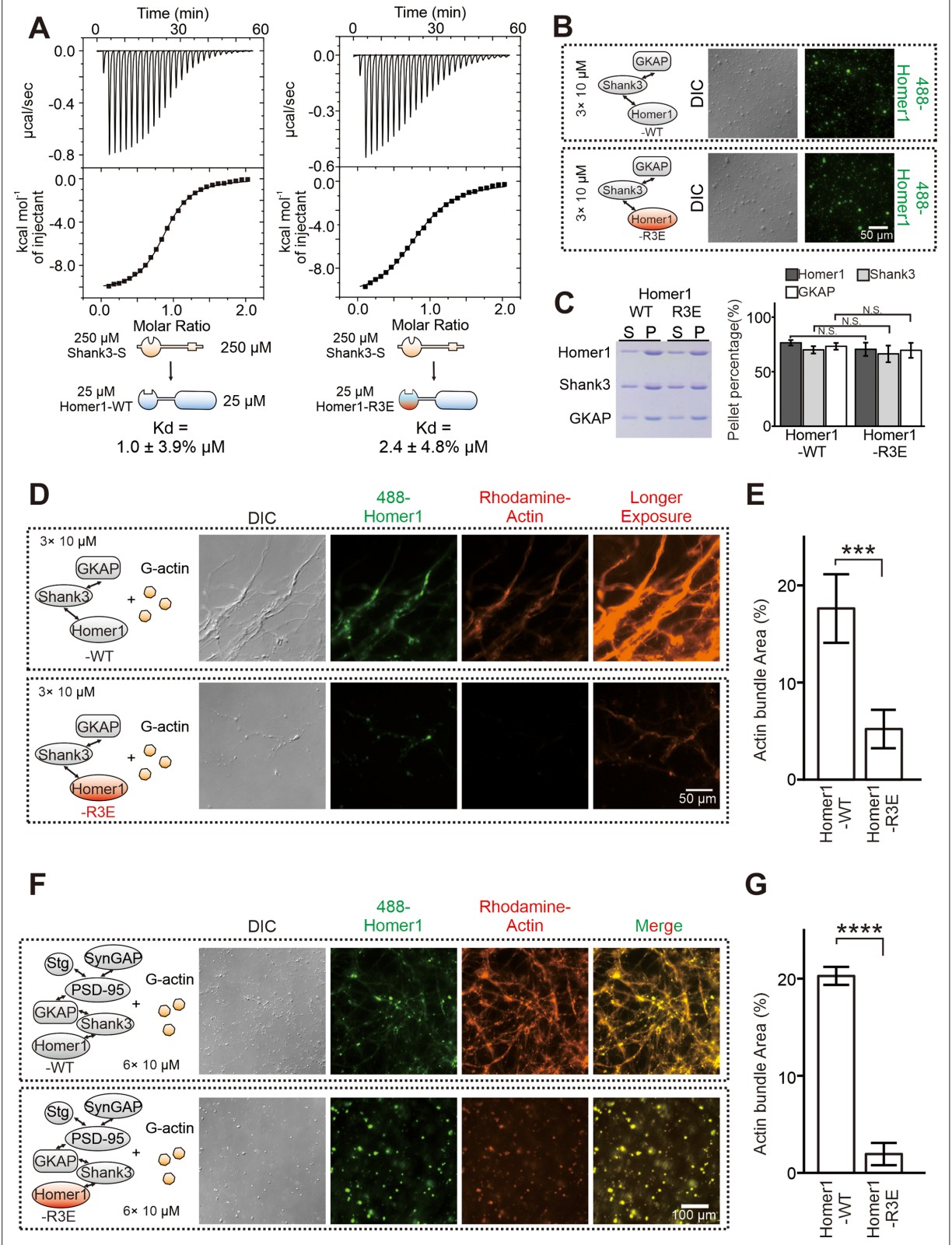

**Figure 3.** The R3E mutation on EVH1 weakens actin bundling without affecting phase separation of Homer with postsynaptic density (PSD) proteins. (**A**) Isothermal titration calorimetry (ITC) measurements showing that Homer1-WT and Homer1-R3E bound to Shank3 with similar affinities. Two hundred and fifty μM Homer1 WT or R3E was titrated to 25 μM Shank3. (**B**) Differential interference contrast (DIC) and fluorescence images showing Homer1-WT or Homer1-R3E (both iFluor-488 labeled) was concentrated into condensates when mixed with Shank3 and GKAP. (**C**) Representative SDS-PAGE

*Figure 3 continued on next page*

*Figure 3 continued*

and quantification data showing the distributions of the GKAP, Shank3, and Homer1 (WT or R3E) recovered in the dilute phase/supernatant (**S**) and condensed phase/pellet (**P**) at indicated protein concentrations. N=6 independent batches of sedimentation assays and are presented as mean ± SD, Student's t-test. NS, not significant. (D and E) DIC/fluorescence images and quantification data showing that actin bundling could be strongly induced by GKAP/Shank3/Homer1-WT condensates, but not by GKAP/Shank3/Homer1-R3E condensates. N=4 independent batches of imaging assays and are presented as mean ± SD, Student's t-test. ***p<0.001. (F and G) DIC/fluorescence images and quantification data showing that actin bundling could be strongly induced by 6× PSD condensates which were enriched with Homer1-WT, but not with Homer1-R3E. Results were from four independent batches of imaging assays and are presented as mean ± SD, Student's t-test. ****p<0.0001.

The online version of this article includes the following source data for figure 3:

**Source data 1.** Original gel image in *Figure 3C*.

the Homer1-R3E-containing 6× PSD condensates were essentially incapable of inducing actin bundle formation, but the Homer1-WT containing 6× PSD condensates could robustly induce actin bundle formation (*Figure 3F and G*). In the Homer1-R3E-containing 6× PSD condensates, actin was largely diffused in solution but with some degree of enrichments in droplets of the PSD condensates (*Figure 3F*). The above results revealed that the Arg residue-containing surface of the Homer EVH1 domain is essential for binding to actin and inducing actin bundle formation. The R3E mutation selectively abolishes the actin bundling capacity of Homer1. Importantly, the R3E mutation neither affects Homer1's binding to Shank3 nor alters the phase separation capacity of Homer1 with PSD proteins.

## Homer1a disrupts the PSD condensate-mediated actin bundling

We next explored whether the PSD condensate-induced actin bundling may be regulated by neuronal activity-related signals. We focused on Homer1a, an alternative splicing isoform of the full-length Homer1. Homer1a lacks the entire C-terminal CC domain and thus the protein is a monomer (*Figure 2A*). *Homer1a* is an early immediate gene and its expression is induced by broad neuronal activities (*Bockaert et al., 2021*). Homer1a can antagonize with the full-length tetrameric Homer proteins by blocking cross-linking of PSD assemblies into large molecular networks or by disrupting connections between the metabotropic glutamate receptors and ionotropic glutamate receptors (*Diering et al., 2017*; *Sala et al., 2003*; *Zeng et al., 2018*). Thus, Homer1a is a versatile and activity-dependent regulator of synaptic plasticity.

We asked whether Homer1a may also antagonize with the full-length Homer1 in bundling actin filaments. We first mixed excess amount of Homer1a (50 μM) with the lower layer of the PSD condensates (each component at 10 μM) before addition of G-actin (also at 5 μM). We found that Homer1a effectively prevented PSD condensate formation and actin bundling (*Figure 4A and B*). We then asked whether pre-formed PSD condensates and resulting bundled actin might be dispersed by addition of Homer1a, an experimental design mimicking Homer1a-induced PSD downscaling in neurons (*Vazdarjanova et al., 2002*). We observed that the Homer1-containing PSD condensates were rapidly dispersed (e.g., from 0 to 20 s in *Figure 4C*) upon addition of Homer1a. Interestingly, the PSD condensate-induced actin bundles were also disassembled, albeit with a time delay (e.g., it took >120 s for the actin bundles to be fully disassembled; *Figure 4C*). Our finding is consistent with a previous discovery showing that increased expression of Homer1a led to dramatic reduction of F-actin in dendritic spines (*Sala et al., 2003*). The observed time delay between the Homer1a-induced synaptic condensate dispersion and actin bundle disassembly is also consistent with our biochemical observations showing that actin bundle formation relies on the formation of the PSD condensates containing the full-length Homer1 or Homer3 (*Figure 1F*). Remarkably, such a time delay was also observed in living neurons showing that de-clustering of synaptic Homer puncta precedes the disassembling of F-actin during activity-dependent synaptic remodeling (*Shiraishi et al., 2003*).

## Phosphorylation on Homer3 weakens the actin bundling by PSD condensates

Homer3 is a unique Homer isoform with its expression relatively restricted to hippocampal CA3 pyramidal neurons and cerebellar Purkinje cells (*Shiraishi et al., 2004*). Three unique CaMKIIα phosphorylation sites have been identified in the central linker region of Homer3 (*Mizutani et al., 2008*; *Figure 2—figure supplement 1B*). Phosphorylated Homer3 displayed preferential localization in the cytosolic fraction, whereas unphosphorylated Homer3 was mainly enriched in the PSD core (*Mizutani*

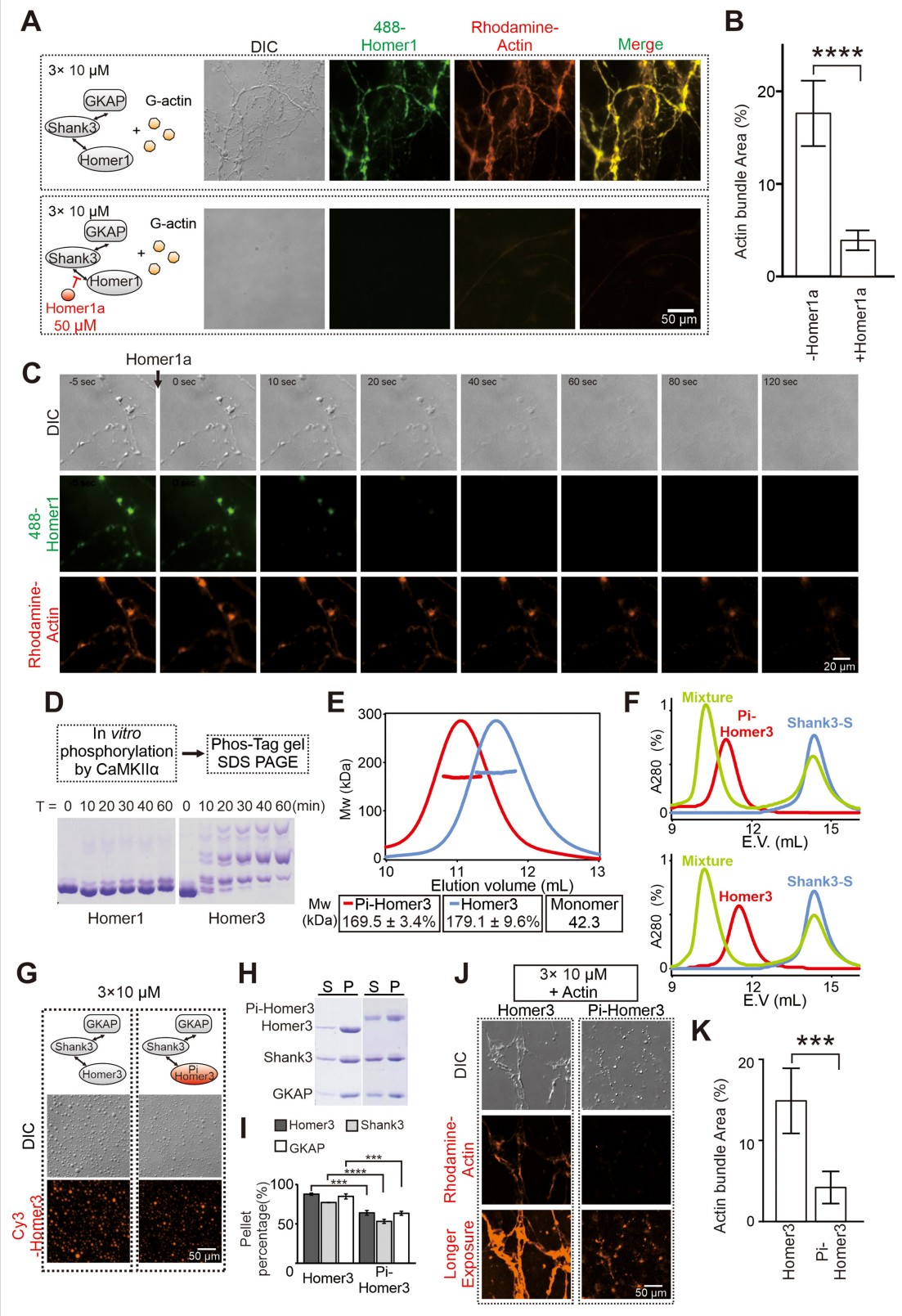

**Figure 4.** Postsynaptic density (PSD) condensate-mediated actin bundling is regulated by Homer1a and CaMKIIα. (**A** and **B**) Differential interference contrast (DIC)/fluorescence images and quantification data showing that pre-adding Homer1a into GKAP/Shank3/Homer1 condensates prevented actin bundle formation. N=4 independent batches for -Homer1a, and 5 for +Homer1a, data are presented as mean ± SD, Student's t-test. ****p<0.0001. (**C**) Time-lapse images showing that adding Homer1a into actin bundles pre-assembled by GKAP/Shank3/Homer1 condensates led to dispersions of

*Figure 4 continued on next page*

*Figure 4 continued*

both PSD condensates and actin bundles. (**D**) Phos-Tag couple with SDS-PAGE assay showing that CaMKIIα could phosphorylate Homer3, but not Homer1 in vitro. (**E**) Fast protein liquid chromatography (FPLC)-SLS analysis showing that phosphorylation of Homer3 did not alter its tetramerization. Homer3 and Pi-Homer3 were assayed in 50 μM. (**F**) FPLC analysis showing that both Homer3 and Pi-Homer3 (50 μM) could interact and form complex with Shank3-dSAM (100 μM). In this assay, the SAM domain on Shank3 was deleted to prevent the phase separation of Homer3 and Shank3 upon mixing. (**G**) DIC and fluorescence images showing that Homer3 and Pi-Homer3 (both Cy3 labeled) was concentrated into condensates when mixed with Shank3 and GKAP. (**H and I**) Representative SDS-PAGE and quantification data showing the distributions of the GKAP, Shank3, and Homer3 (or Pi-Homer3) recovered in the dilute phase/supernatant (**S**) and condensed phase/pellet (**P**) at indicated protein concentrations. N=3 independent batches of sedimentation assays and are presented as mean ± SD, Student's t-test. \*\*\*p<0.001, \*\*\*\*p<0.0001. (**J and K**) DIC/fluorescence images and quantification data showing that actin bundling could be strongly induced by 3× PSD condensates enriched with Homer3, but not Pi-Homer3. Results were from three independent batches of imaging assays and are presented as mean ± SD, Student's t-test. \*\*\*p<0.001.

The online version of this article includes the following source data for figure 4:

**Source data 1.** Original gel images presented in *Figure 4*.

**Source data 2.** Original gel images presented in *Figure 4*.

---

*et al., 2008*). Under synaptic depolarization, NMDAR-mediated Ca²⁺ influx triged Homer puncta de-clustering, resulting in translocation of Homer3 from dendritic spines to nearby shafts (*Guo et al., 2015*; *Okabe et al., 2001*; *Shiraishi et al., 2003*). Interestingly, Homer3 with mutations blocking CaMKIIα phosphorylation failed to undergo Ca²⁺ influx-induced spine-to-shaft dispersion (*Guo et al., 2015*).

We next used our in vitro reconstitution system to evaluate CaMKIIα-mediated phosphorylation of Homer3 on its actin bundling. We first verified using in vitro phosphorylation assay that Homer3, but not Homer1, is a good substrate of CaMKIIα. In the presence of 0.5 μM of purified active CaMKIIα, Homer3 (at 20 μM) was rapidly and near completely phosphorylated within 60 min. Under the same condition, only a small fraction of Homer1 was phosphorylated (*Figure 4D*). We separated the phosphorylated Homer3 (Pi-Homer3) from CaMKIIα by size exclusion column. Fast protein liquid chromatography (FPLC) coupled with static light scattering experiment showed that phosphorylation of Homer3 neither changed its tetramerization state, nor affected its interaction with Shank3 (*Figure 4E and F*). Pi-Homer3 also underwent phase separation with Shank3 and GKAP, with a slightly weaker capacity than unphosphorylated Homer3 (*Figure 4G–I*). Remarkably, the lower layer PSD condensates containing Pi-Homer3 exhibited a dramatically weakened capacity in inducing actin bundle formation (*Figure 4J and K*). It is possible that, upon synaptic stimulation, Ca²⁺ influx and subsequent CaMKIIα-mediated phosphorylation of Homer3 may first locally disassemble the PSD-tethered actin bundles for the subsequent remodeling of the synaptic actin cytoskeletal structures and expression of synaptic plasticity.

## Homer promotes cell migration via binding to actin

Besides the nerve system, Homer proteins also play diverse roles in cells of other tissues. For example, Homer proteins are expressed in skeletal muscles and regulate muscle differentiation and regeneration (*Bortoloso et al., 2006*; *Salanova et al., 2002*; *Stiber et al., 2005*). Homer1 is also required for vascular smooth muscle cell migration and proliferation during development (*Jia et al., 2017*). In neutrophil-like HL-60 cell, Homer3 is essential for polarity and migration of neutrophils via regulating actin cytoskeletons (*Wu et al., 2015*).

We next evaluated the role of the Homer EVH1-mediated actin interaction using transwell cell migration assay (*Figure 5A–C*). Consistent with the findings in the literature (*Wu et al., 2015*), overexpression of Homer1 (tagged with mCherry at the C-terminus and termed as Homer1-WT-mCherry) in Hela cells resulted in a higher level of cell motility compared with the mCherry control. In contrast, Homer1-R3E-mCherry had a significantly reduced activity in promoting cell migration. Interestingly, regaining the actin-binding capacity Homer1-R3E by adding the EVH1 domain from Enah to Homer1-R3E (see *Figure 5A* for the design of the construct; *Figure 2—figure supplement 1C* for the properties of the EVH1 domains) converted the Homer1 mutant to be able to promote cell migration (*Figure 5B and C*). We further biochemically verified that both Homer EVH1 and Enah EVH1 domain, in their tetramer state, could directly bind to and co-sediment with F-actin in a high-speed-centrifugation-based F-actin-binding assay (*Figure 5D*). It should also be noticed that in our cell migration assay, we introduced a W23A mutation to the Enah EVH1 domain to disrupt its binding to

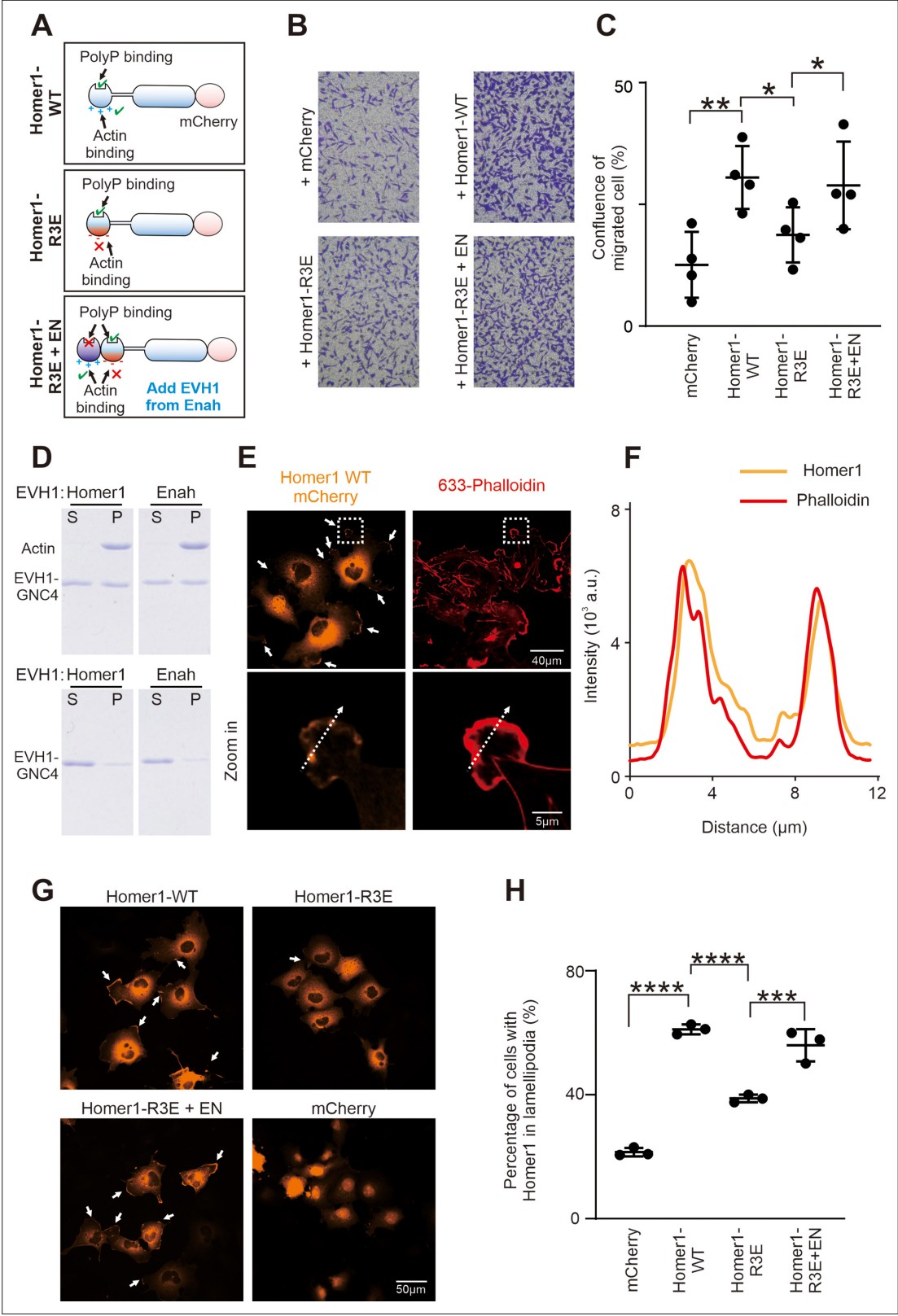

**Figure 5.** Homer binding to F-actin promotes cell migration. (**A**) Schematic diagram showing Homer1-WT-mCherry, Homer1-R3E-mCherry, and Homer1-R3E+EN-mCherry constructs designed for cell migration and cellular actin-binding assays. (**B**) Transwell migration assay measuring the cell migration activities of HeLa cells transfected with mCherry, Homer1-WT-mCherry, Homer1-R3E-mCherry, and Homer1-R3E+EN-mCherry. (**C**) Quantification of cell migration activities as described in (**B**). Results were from four independent batches of migration assays and are presented as mean ± SD. **p<0.01,

*Figure 5 continued on next page*

*Figure 5 continued*

*p<0.05, using one-way ANOVA with Tukey's multiple comparisons test. (**D**) SDS-PAGE showing Homer1 EVH1 and Enah EVH1 co-sedimented with F-actin in the high-speed centrifugation-base actin-binding assay. Top row: with actin added; bottom row: without actin added. (**E**) Representative fluorescence images showing the localizations of Homer1-WT-mCherry in COS-7 cell line. Lamellipodia F-actin bundles were stained by Alexa-633-phalloidin. (**F**) Line-plots showing fluorescence intensity of Homer1-WT-mCherry and lamellipodia F-actin in a COS-7 cell as indicated in panel E. (**G–H**) Representative microscopic images and quantification results showing the lamellipodia localization of Homer1-WT-mCherry, Homer1-R3E-mCherry, Homer1-3E+EN-mCherry, and mCherry in COS-7 cells. N=3 independent batches of imaging assay, data are presented as mean ± SD. ***p<0.001, ****p<0.0001, using one-way ANOVA with Tukey's multiple comparisons test.

The online version of this article includes the following source data for figure 5:

**Source data 1.** Original gel images presented in *Figure 5*.

**Source data 2.** Original gel images presented in *Figure 5*.

PRM motif-containing targets, so that the rescue of the cell migration by Homer1-R3E+EN-mCherry is solely due to the regain of its actin-binding capability (*Figure 5A*).

Consistent with the above cell migration experiment, Homer1-mCherry expressed in COS-7 cells showed prominent localizations at lamellipodia, which are membrane protrusions enriched with bundled actins and critical for cell migrations. Staining of Homer1-mCherry expressed cells with Alexa-633 phalloidin revealed colocalizations between F-actin and Homer1-mCherry at lamellipodia (*Figure 5E–F*). The lamellipodia localization of Homer1-mCherry was specific as expressed mCherry tag was not localized in lamellipodia, but mainly diffused in cytosol and enriched in nuclei instead (*Figure 5G*). The lamellipodia localization of Homer1 was significantly impaired by the R3E mutation of Homer1 ('Homer1-R3E' in *Figure 5G–H*). Importantly, the disrupted lamellipodia localization of the R3E mutation could be rescued by inserting the actin-binding EVH1 domain of Enah (i.e., the 'Homer-3E+EN' chimera) (*Figure 5G–H*).

## Actin binding is important for the synaptic localization and functions of Homer

Finally, we studied the physiological relevance of Homer-mediated actin binding in neurons by using Homer1-WT, Homer1-R3E, and Homer1-R3E+EN constructs. We first compared the synaptic targeting of all three Homer constructs exogenously expressed in cultured mice hippocampal neurons. Homers are highly enriched in dendritic spines in neurons according to previous studies (*Sala et al., 2001*). Overexpressed Homer1-WT-mCherry also displayed a strong synaptic localization in cultured hippocampal neurons, as the averaged spine-to-shaft ration of Homer1-WT-mCherry was significantly higher than mCherry (*Figure 6A–C*). In contrast, Homer1-R3E-mCherry mutant showed a remarkably reduced synaptic enrichment compared with Homer1-WT (*Figure 6A–C*). Notably, such decreased synaptic localization of Homer1-R3E-mCherry protein was fully rescued by Homer1-R3E+EN-mCherry (*Figure 6A–C*). In conclusion, we conclude that the actin binding of Homer is essential for the synaptic targeting of Homer. Our data also correlate well with the finding that phosphorylation of Homer3 by CaMKIIα, which results in a weakened actin budling ability of the protein, also exhibits reduced spine localizations in neurons (*Guo et al., 2015*; *Okabe et al., 2001*; *Shiraishi et al., 2003*).

Next, we investigated whether the actin binding of Homer contributes to the development of dendritic spine structures. Overexpression of PSD scaffolds like Shank and Homer proteins is known to induce spine enlargement (*Sala et al., 2001*; *Yoon et al., 2021*). We then asked whether spine enlargement induced by Homer requires the specific Homer-actin interaction. As expected, overexpression of Homer1-WT-mCherry in cultured mice hippocampal neurons significantly increased the widths of spine heads (*Figure 6A and D*). In contrast, overexpression of the Homer-R3E mutant did not increase the widths of the spine heads, but instead slightly reduced the spine head size when compared to the mCherry control possibly due to the dominant negative effect of the mutant. Importantly, the Homer1-R3E+EN-mCherry protein, much like Homer1-WT-mCherry, also induced spine head enlargement (*Figure 6A and D*). Thus, we conclude that the EVH1-mediated interaction between Homer and actin is critical for Homer-induced spine maturation.

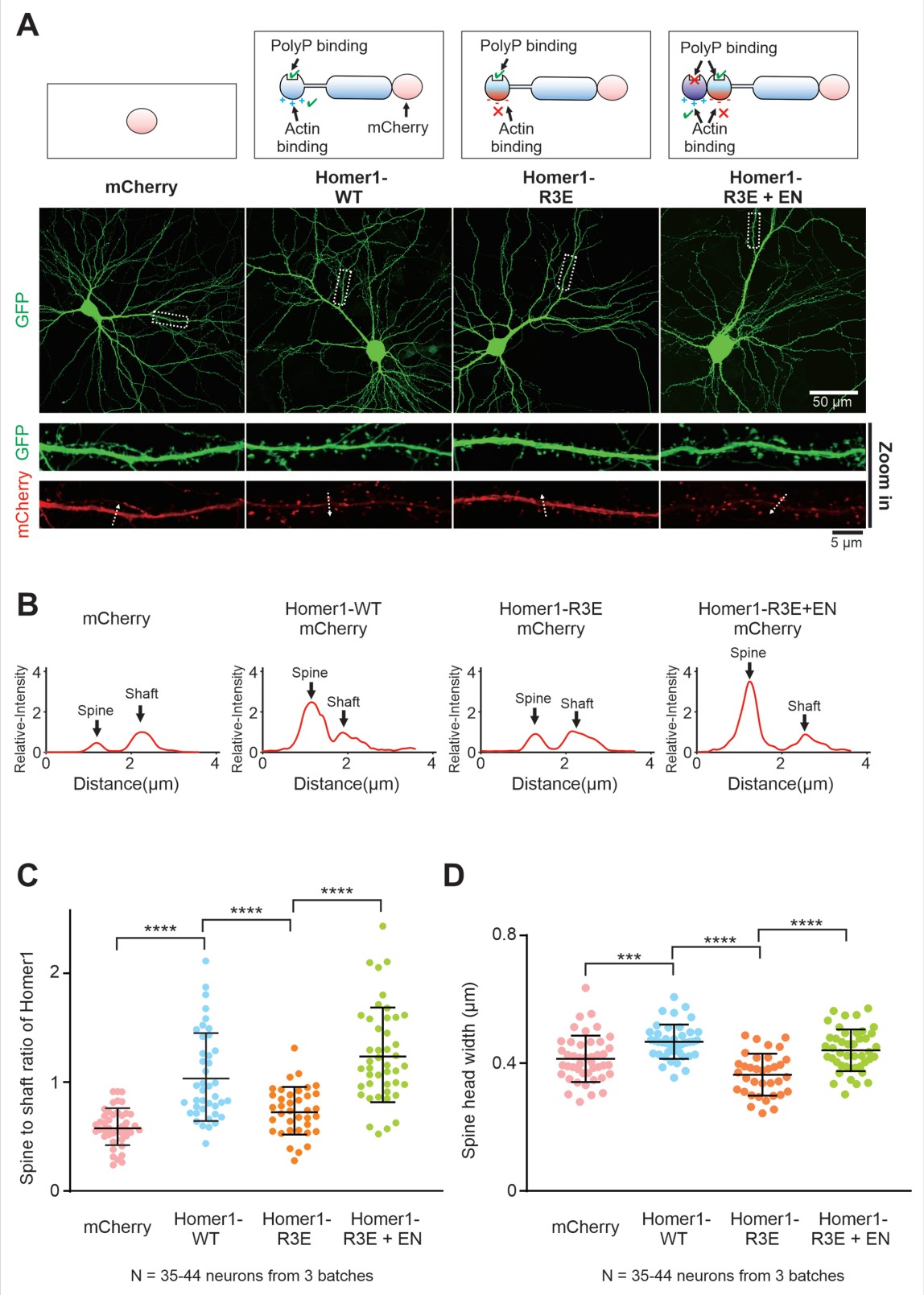

**Figure 6.** Actin binding is important for the synaptic localization and functions of Homer. (**A**) Cultured mice hippocampal neurons transfected with mCherry, Homer1-WT-mCherry, Homer1-R3E-mCherry, and Homer1-R3E+EN-mCherry at 14 days in vitro (DIV). GFP was co-transfected with these constructs as the cell fill. After 7 days of expression, neurons were fixed and mounted for imaging by confocal microscope. (**B**) The fluorescence profiles of the mCherry constructs across the lines indicated in (**A**). All fluorescence intensities along the line are normalized to the peak intensity of shaft.

*Figure 6 continued on next page*

*Figure 6 continued*

(**C**) Quantification of averaged spine/shaft intensity ratios of mCherry and Homer-mCherry variants in (**A**) and (**B**). N represents the cell numbers used in each quantification. Three independent batches of cultures were imaged for each group for quantification. Error bars indicate ± SD. ***$p < 0.001$, ****$p < 0.0001$. One-way ANOVA with Tukey's multiple comparisons test. (**D**) Quantification of image data in (**A**) showing increased spine head width in neurons expressing Homer1-WT-mCherry and Homer1-R3E+EN-mCherry, and decreased spine head width in neurons expressing Homer1-R3E. N represents the cell numbers used in each quantification. Three independent batches of cultures were imaged for each group for quantification. Error bars indicate ± SD. ***$p < 0.001$, ****$p < 0.0001$. One-way ANOVA with Tukey's multiple comparisons test.

## Discussion

The most salient finding of the current study is the direct communication between the PSD condensates and actin cytoskeleton. We found that actin polymerization can nucleate at PSD condensates and subsequently bundled actin can emanate from PSD condensates. The communication between PSD condensates and actin cytoskeleton requires direct binding between Homer and actin as well as Homer to form condensed molecular assembly with other PSD scaffold proteins. The direct communication between the PSD condensates and the actin cytoskeleton may have multiple implications on the structural plasticity of synapses. An enlarged PSD condensate can promote F-actin network formation; in return a more exuberant actin cytoskeleton may provide structural support to stabilize an enlarged PSD and thus support structural stability of a potentiated synapse. Conversely, shrinking of the PSD in a de-potentiated synapse can cause destabilization of the synaptic actin network; in return the weakened actin cytoskeleton further destabilizes the PSD and eventually leads to overall spine shrinkage. Most likely, numerous actin regulatory proteins existing in dendritic spines may either work together with the PSD condensates or in distinct processes in modulating spine actin cytoskeleton network.

We found that the Homer-containing PSD condensates, instead of Homer alone, can promote actin bundle formation. This discovery implies that the synaptic actin network can be modulated by targeting PSD condensates without directly interfering Homer/actin interaction. Indeed, the Homer1a can effectively disperse the PSD condensates and destabilize already formed bundled actin filaments. Homer1a is the gene product of the alternatively splice isoform of *Homer1c* lacking the C-terminal CC domain. Thus, although the bindings to its targets including actin and Shank is not affected by the alternative splicing, the monomeric Homer1a can sensitively modulate PSD condensate formation in a dose-dependent manner (*Figure 4*; *Zeng et al., 2018*). The expression level of Homer1a is tightly coupled to synaptic activities and increase of Homer1a expression is known to be inversely correlated with the downscaling of synapses as well as PSD sizes in neurons (*Sala et al., 2003*). Our study in *Figure 4* showed that Homer1a can rapidly disperse PSD condensates. With a delay, Homer1a-mediated PSD condensate dispersion leads to destabilization of bundled actin, indicating that the bundled actin filaments are indeed stabilized by PSD condensates. Many factors can regulate PSD condensate formation or dispersion (*Chen et al., 2020*; *Zeng et al., 2018*). Therefore, the PSD condensates may serve as a hub to connect different synaptic regulatory signals to the spine actin cytoskeletons.

CaMKIIα-mediated phosphorylation of Homer3, but not the other isoforms of Homer, is another unique mechanism underlying the regulated communication between PSD condensates and actin cytoskeleton. It has been observed that synaptic stimulation induces rapid CaMKIIα activation (within a few minutes) accompanied by transient disassembly of actin filaments in dendritic spines (*Basu and Lamprecht, 2018*; *Dillon and Goda, 2005*; *Lamprecht, 2014*; *Lei et al., 2016*). Our study revealed that CaMKIIα phosphorylates specific Ser residues in the central linker region immediately following the EVH1 domain. Introduction of negative charges near the basic charged EVH1 domain by the phosphorylation disrupts the Homer3/actin binding. The net consequence of CaMKIIα phosphorylation of Homer3 in PSD is detachment of actin filaments from the PSD condensates and subsequent increase of actin cytoskeletal dynamics, a step critical for activity-induced synaptic remodeling and long-term stabilization. CaMKIIα and -β are both enriched in synapses and can form hetero-dodecamers. The two isoforms of the kinase share very similar catalytic activities toward their substrates, but CaMKIIβ in its active state can also bind to and bundle actin filaments. $Ca^{2+}$-induced autophosphorylation of CaMKIIβ disrupts its actin binding and bundling activity. Thus, under the resting condition, CaMKIIβ and Homer3 can both bind to actin filaments. Upon stimulation, $Ca^{2+}$ influx can lead to rapid Homer phosphorylation by CaMKII as both the kinase and Homer3 are anchored on actin filaments.

In summary, we discovered in this study that the PSD condensates and actin cytoskeleton, two major synaptic molecular assemblies formed via phase separation/transition, directly communicate with each other, forming a mutually reinforcing system capable of modulating synaptic plasticity. The interaction between the PSD condensates and actin cytoskeleton can be modulated by elements such as CaMKII and the immediate early gene product *Homer1a*, both of which are key synaptic plasticity regulators.

## Materials and methods

### Protein expression and purification

Sequences coding various proteins were generated using standard PCR-based methods, each cloned into a vector containing an N-terminal Trx-His$_6$ or a His$_6$-affinity tag followed by an HRV 3C cutting site. All constructs were confirmed by DNA sequencing. Recombinant proteins were expressed in *Escherichia coli* BL21-CodonPlus (DE3)-RIL cells (Agilent) in LB medium at 16°C overnight and protein expression was induced by 0.25 mM IPTG (final concentration) at OD$_{600}$ between 0.6 and 0.8. Protein purification was performed as described previously (***Zeng et al., 2018***). Typically, each recombinant protein (PSD-95, SynGAP, GKAP, Shank3, and calmodulin) was purified using a nickel-NTA agarose affinity column followed by a size-exclusion chromatography (Superdex 200 or Superdex 75, GE Healthcare) with a column buffer containing 50 mM Tris, pH 8.0, 100 mM NaCl, 2 mM DTT. After cleavage by HRV 3C protease, the His$_6$-affinity or Trx-His$_6$ tag was separated by another step of size-exclusion chromatography using Superdex 200 or Superdex 75 with the column buffer containing 50 mM Tris, pH 8.0, 100 mM NaCl, 2 mM DTT.

For purifications of Homer1 and Homer3, a mono Q ion-exchange chromatography (GE Healthcare) was added to remove DNA contamination and His$_6$-tag after the HRV 3C protease cleavage. Protein was exchanged into a buffer containing 50 mM Tris, pH 8.0, 100 mM NaCl, 2 mM DTT by HiTrap desalting column (GE Healthcare).

Stg was produced as described previously (***Zeng et al., 2019***). Stg was expressed at 37°C for 3 hr to minimize protein degradation. Proteins eluted from nickel-NTA agarose affinity column were then purified by Superdex 75 size-exclusion chromatography with a column buffer containing 50 mM Tris, pH 8.0, 300 mM NaCl, 2 mM DTT. After affinity tag cleavage by HRV 3C protease, a mono S ion-exchange chromatography (GE Healthcare) was used to remove the Trx-His$_6$ tag from Stg. Protein was exchanged into a buffer containing 50 mM Tris, pH 8.0, 100 mM NaCl, 2 mM DTT by a HiTrap desalting column.

Rat CaMKIIα kinase domain 1–314 and mouse CaMKIIβ holoenzyme were generated as described before (***Cai et al., 2021***). Kinases were co-expressed with $\lambda$ phosphatase in *E. coli* BL21-CodonPlus (DE3)-RIL cells. CaMKIIα kinase domain was purified using a nickel-NTA agarose affinity column followed by a size-exclusion chromatography (Superdex 75) with a column buffer containing 50 mM Tris, pH 8.0, 100 mM NaCl, 1 mM EDTA, 2 mM DTT, and 10% glycerol. CaMKIIβ was purified using a nickel-NTA agarose affinity column followed by a size-exclusion chromatography (Superdex 200) with a column buffer containing 50 mM Tris, pH 8.0, 200 mM NaCl, 1mM EDTA, 5 mM DTT, and 10% glycerol. After HRV 3C cleavage, the sample was loaded onto a Mono Q column and eluted by a NaCl gradient. The eluted protein was then loaded onto a Superose 6 10/300 gel filtration column in a buffer containing 50 mM Tris, pH 8.0, 200 mM NaCl, 1mM EDTA, 5 mM DTT, and 10% glycerol.

### Protein fluorescence labeling

For amide labeling: Highly purified proteins were exchanged into a NaHCO$_3$ buffer (containing 100 mM NaHCO$_3$ pH 8.3, 300 mM NaCl, 1 mM EDTA, and 2 mM DTT) and concentrated to 5–10 mg/mL. Alexa-647 NHS ester (Invitrogen) or iFluor-488/Cy3 NHS ester (AAT Bioquest) were dissolved by DMSO making stock solutions at the concentration of 10 mg/mL. Each dye and the protein to be labeled were mixed at a molar ratio of 1:1 and the reaction was lasted for 1 hr at room temperature. Reaction was quenched by 200 mM Tris, pH 8.2. The fluorophores and other small molecules were removed from the proteins by passing the reaction mixture through a HiTrap desalting column with buffer containing 50 mM Tris, pH 8.0, 100 mM NaCl, and 2 mM DTT.

Fluorescence labeling efficiency was measured by Nanodrop 2000 (Thermo Fisher). In imaging assays, fluorescence-labeled proteins were further diluted with the corresponding unlabeled proteins in the same buffer. Dilution ratio was specified in the legend of each figure.

### Actin preparation

Actin and rhodamine-labeled actin (Cytoskeleton, Inc) were first dissolved in 5 mM Tris-HCl pH 8.0, 0.2 mM $CaCl_2$ on ice for 1 hr. Actin was centrifuged at 16,873 g for 15 min at 4°C. Rhodamine-actin was centrifuged at 100,000 × $g$ at 4°C for 10 min. The supernatants containing soluble actin was collected. The final concentration of actin or rhodamine-actin was set at 20 µM.

### Imaging-based assay of phase separation

Imaging-based phase separation assays followed our previously described procedures (*Zeng et al., 2018*; *Zeng et al., 2019*). Briefly, proteins (with affinity tags cleaved and removed) were prepared in a buffer containing 50 mM Tris, pH 8.0, 100 mM NaCl, 0.5 mM ATP, 1 mM $MgCl_2$, and 2 mM DTT, and pre-cleared via high-speed centrifugations. Proteins were then mixed or diluted with buffer to designated combinations and concentrations.

For phase separation-mediated actin bundling assay, protein samples were first mixed in 37.5 µL buffer before adding actin. The addition of 12.5 µL 20 µM actin leading to a final actin concentration at 5 µM. For phase separation assays without actin, 12.5 µL actin storing buffer (5 mM Tris-HCl pH 8.0, 0.2 mM $CaCl_2$) was added into a 37.5 µL protein condensates. The mixture was gently pipetted no more than twice to avoid the twisting of assembled F-actin. The protein mixture was placed at room temperature for 10 min before it was injected into a homemade flow chamber for DIC and fluorescent imaging with a Nikon Ni-U upright fluorescence microscope (20× and 40× lenses) or with a Zeiss LSM880 confocal microscope (63× lens) imaging. Images were analyzed by the ImageJ software.

For sedimentation assays, typically, the final volume of each reaction is 50 µL. After 10 min equilibrium at room temperature, protein samples were subjected to sedimentation at 16,873 × $g$ for 10 min at 25°C on a table-top temperature-controlled micro-centrifuge. After centrifugation, the supernatant and pellet were immediately separated into two tubes. The pellet fraction was thoroughly re-suspended with the same 50 µL buffer. Proteins from both fractions were analyzed by SDS-PAGE with Coomassie blue staining. Band intensities were quantified using the ImageJ software.

### F-actin-binding assay

Actin at 10 µM was first polymerized in 50 mM Tris, pH 8.0, 50 mM KCl, 2 mM $MgCl_2$ 1 mM ATP, and 2 mM DTT for 1 hr, followed by addition of 10 µM EVH1 proteins. F-actin and EVH1 proteins were incubated for 30 min before being centrifuged at 100,000 × $g$ for 1 hr. Proteins from both pellet and supernatant fractions were analyzed by SDS-PAGE with Coomassie blue staining. Actin polymerization, EVH1 protein incubation, and high-speed centrifugation were all performed at room temperature.

### ITC assay

ITC measurements were carried out on a MicroCal VP-ITC calorimeter at 25°C. Proteins used for ITC measurements were dissolved in an assay buffer composed of 50 mM Tris, pH 8.0, 100 mM NaCl, 1 mM EDTA, and 2 mM DTT. Affinity tags on proteins were cleaved and removed. High concentration of protein was loaded into the syringe and titrated into the cell containing low concentration of corresponding interactors (concentrations for each reaction are indicated in the figure legends). For each titration point, a 10 µL aliquot of a protein sample in the syringe was injected into the interacting protein in the cell at a time interval of 2 min. Titration data were analyzed using the Origin7.0 software and fitted with the one-site binding model.

### CaMKIIα phosphorylation assay

CaMKIIα kinase domain (100 µM) was first mixed with 200 µM calmodulin for auto-phosphorylation in a buffer containing 50 mM Tris pH 8.0, 100 mM NaCl, 10 mM ATP, 5 mM $CaCl_2$, 2 mM DTT, 5 mM $MgCl_2$ and 10% glycerol at room temperature for 10 min. Homer protein at 20 µM was mixed with 0.5 µM auto-phosphorylated CaMKIIα kinase domain in a reaction buffer containing 50 mM Tris pH 8.0, 100 mM NaCl, 10 mM ATP, 2 mM DTT, and 5 mM $MgCl_2$ at room temperature overnight. To

remove the enzymes after phosphorylation, samples were loaded into size-exclusion chromatography Superdex 200 with buffer containing 50 mM Tris, pH 8.0, 100 mM NaCl, 2 mM DTT.

## FPLC coupled with static light scattering

The analysis was performed on an Agilent InfinityLab system coupled with a static light scattering detector (miniDawn, Wyatt) and a differential refractive index detector (Optilab, Wyatt). Protein samples with indicated concentrations were loaded into a Superose 12 10/300 GL column (GE Healthcare) pre-equilibrated with 50 mM Tris, pH 8.0, 100 mM NaCl, 1 mM EDTA, 1 mM DTT buffer. Data were analyzed using ASTRA 6 software (Wyatt).

## Cell migration assay

Hela and COS-7 cells used in this study were purchased from ATCC and used without further authentication. The cells were tested without mycoplasma contamination. Hela Cells were individually transfected with mCherry or mCherry-tagged Homer1 (WT, R3E, and R3E-ENAH) by electroporation (Nucleofector Kit T). Cell migration experiment was performed using Transwell membrane filter inserts (8 mm pore size, Corning costar). $5 \times 10^4$ HeLa cells were seeded into the upper chamber and allowed to migrate into the lower chamber for 16–18 hr at 37°C. Cells in the upper chamber were carefully wiped by cotton buds, cells at the bottom of the membrane were washed once with PBS, and fixed by 4% (vol/vol) paraformaldehyde (PFA) together with 4% (wt/vol) sucrose in PBS (pH 7.5) and then stained with Crystal Violet Staining Solution (Beyotime Biotechnology). The confluence level of migrated cells was imaged under a light microscope from five random fields of each well. Statistical data was obtained from four independent experiments.

## Lamellipodia localization assay

COS-7 cells were cultured on gelatin-coated coverslips in 12-well plates. Transfection was performed when the cell confluence reached 40%. COS-7 cells were individually transfected with mCherry or mCherry-tagged Homer1 (WT, R3E, and R3E-ENAH) by ViaFect Transfection Reagent (Promega). Briefly, 1 µg DNA and 3 µL transfection reagent were mixed in 100 µL Opti-MEM media (GIBCO) for 20 min at room temperature before the mixture was added into each well. Seventeen hours after transfection, cells were fixed with 4% (vol/vol) PFA together with 4% (wt/vol) sucrose in PBS (pH 7.5).

For F-actin staining, fixed cells were first permeabilized with 0.2% Triton X-100 in PBS (pH 7.5) for 20 min, and then blocked by blocking buffer containing 5% (wt/vol) BSA in PBS (pH 7.5) for 2 hr at room temperature. After blocking, cells were stained with Alexa-633 phalloidin (Thermo Fisher, 1:1000 in blocking buffer) for 1 hr at room temperature. Mounted cells were imaged at Nikon Ni-U upright fluorescence with a 40× lens. The percentage of cells with mCherry signal in lamellipodia were quantified by ImageJ. Statistical data was plotted from three independent batches of cells with >300 cells for each batch in a blinded manner.

## Primary hippocampal neuron culture

Hippocampal neuronal cultures were prepared from E17 C57BL/6 WT mice hippocampi. Cells were seeded on PDL/laminin double-coated glass coverslips (Neuvitro) in 12-well plates. The cells were plated in neurobasal media containing 50 U/mL penicillin, 50 mg/mL streptomycin, and 2 mM GlutaMax supplemented with 2% (vol/vol) B27 (GIBCO) and 10% FBS. After an overnight plating, cells were cultured in neurobasal media supplemented with 50 U/mL penicillin, 50 mg/mL streptomycin, 2 mM GlutaMax, 2% (vol/vol) B27 (GIBCO), and 1% FBS. At DIV9, cells were maintained in neurobasal media with 2 mM GlutaMax, 2% B27, 1% FBS, 1× FDU. Cells were cotransfected at DIV14 with plasmids by using Lipofectamine 2000 reagent (Invitrogen). Cells were fixed at DIV21 with 4% (vol/vol) PFA together with 4% (wt/vol) sucrose in 1× PBS (pH 7.5) and then mounted on slides for imaging.

## Quantification and statistical analysis

Statistical parameters including the definitions and exact values of n (e.g., number of experiments, number of spines, number of cells, etc.), distributions and deviations are reported in the figures and corresponding figure legends. Data of in vitro phase separation imaging assay expressed as mean ± SD. Data of primary mice neuron culture were expressed as mean ± SD; NS, not significant, *p<0.05,

**p<0.01, ***p<0.001, and ****p<0.0001 using Student's t-test or one-way ANOVA with Tukey's multiple comparison test.

Data are judged to be statistically significant when p<0.05 by one-way ANOVA with Tukey's multiple comparison test. None of the data were removed from our statistical analysis as outliers. Statistical analysis was performed in GraphPad Prism. All experiments related to cell cultures and imaging studies were performed in blinded fashion.

## Acknowledgements

This work was supported by a grant from National Science Foundation of China (82188101), a grant from the Minister of Science and Technology of China (2019YFA0508402), a grant from Shenzhen Bay Laboratory (S201101002), Shenzhen Talent Program (KQTD20210811090115021), Shenzhen Science and Technology Basic Research Program (JCYJ20220818100215033), Guangdong Innovative and Entrepreneurial Research Team Program (2021ZT09Y104), grants from Research Grant Council of Hong Kong (AoE-M09-12, 16104518, and 16101419), and an HFSP Research Grant (RGP0020/2019) to MZ.

## Additional information

### Competing interests

Mingjie Zhang: Reviewing editor, *eLife*. The other authors declare that no competing interests exist.

### Funding

| Funder | Grant reference number | Author |
| --- | --- | --- |
| National Natural Science Foundation of China | 82188101 | Mingjie Zhang |
| Shenzhen Bay Laboratory | S201101002 | Mingjie Zhang |
| Guangdong Province Introduction of Innovative R&D Team | 2021ZT09Y104 | Mingjie Zhang |
| Research Grants Council, University Grants Committee | AoE-M09-12 | Mingjie Zhang |
| Human Frontier Science Program | RGP0020/2019 | Mingjie Zhang |
| Ministry of Science and Technology of the People's Republic of China | 2019YFA0508402 | Mingjie Zhang |
| Research Grants Council, University Grants Committee | 16104518 | Mingjie Zhang |
| Research Grants Council, University Grants Committee | 16101419 | Mingjie Zhang |
| Shenzhen Bay Laboratory | S201101002 | Mingjie Zhang |
| Shenzhen Talent Program | KQTD20210811090115021 | Mingjie Zhang |
| Shenzhen Science and Technology Basic Research Program | JCYJ20220818100215033 | Mingjie Zhang |

The funders had no role in study design, data collection and interpretation, or the decision to submit the work for publication.

## Author contributions
Xudong Chen, Data curation, Formal analysis, Validation, Investigation, Methodology, Writing - original draft, Writing - review and editing; Bowen Jia, Formal analysis, Investigation; Shihan Zhu, Investigation; Mingjie Zhang, Conceptualization, Resources, Supervision, Funding acquisition, Writing - original draft, Project administration, Writing - review and editing

## Author ORCIDs
Xudong Chen ⓘ http://orcid.org/0000-0002-9433-3732
Mingjie Zhang ⓘ http://orcid.org/0000-0001-9404-0190

## Decision letter and Author response
Decision letter https://doi.org/10.7554/eLife.84446.sa1
Author response https://doi.org/10.7554/eLife.84446.sa2

## Additional files

### Supplementary files
• MDAR checklist

• Source data 1. The source data contain the raw numeric data for plotting bar graphs presented in the paper.

### Data availability
Source data provided for all gel images (both raw unlabeled full gels and annotated full gels as well as Excel data files for all bar graphs).

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
