## [Editor Report]

This paper used a reconstitution approach to show that in vitro reconstituted PSD condensates can promote actin polymerization and filamentous actin bundling in the absence of other actin regulatory proteins. The authors further show that the EVH1 domain of Homer is responsible for this activity, which is also regulated by CaMKII. Together, this convincing evidence provides the fundamental insight that the crosstalk between PSD and the spine cytoskeleton may be modulated by targeting the phase separation of the PSD condensates.

---

## [Decision Letter]

**Decision letter after peer review:**

Thank you for submitting your article "Phase separation-mediated actin bundling by the postsynaptic density condensates" for consideration by *eLife*. Your article has been reviewed by 2 peer reviewers, and the evaluation has been overseen by a Reviewing Editor and Richard Aldrich as the Senior Editor. The following individual involved in the review of your submission has agreed to reveal their identity: Dragomir Milovanovic (Reviewer #2).

Essential revisions:

There are several issues that can be further improved by additional control experiments or discussions on the shortcomings of the current form (if you do not want to provide further experimental evidence).

1) Quantitative assessment of in vitro experiments, statistical analyses: several of the in vitro experiments might gain from statistical analyses; examples are Figures 3E, 3G, 4B, 4I, and 4K. These experiments are used to establish effects, it is unclear why they are not statistically analyzed for the presence of differences. If the authors think that it is inappropriate to do so, it should be stated why and a word of caution on the conclusions may be appropriate.

2) Localization and roles of Homer proteins in neurons: the analyses of the experiments in Figure 6 seem preliminary. It is shown that transfected Homer1 (WT, R3E, R3E+EN) induces changes in spine width monitored by co-transfected GFP. R3E disrupts the ability of Homer1 to enhance spine head width, and restoring actin binding via the addition of EN restores it partially. It is important to assess whether the three Homer versions used in this experiment are expressed at similar levels and localized similarly. From the presented data, it seems that these analyses should be possible without new experiments. From the sample images, it seems that the R3E mutant Homer1 does not localize to spines, while the other versions do. The interpretation of this experiment is very different if actin-binding of Homer1 is necessary for the delivery of Homer1 to spines vs. Homer1 is localized to spines independent of its ability to bind to actin, but spine head width is controlled by Homer's ability to bind to actin.

3) It is well established that molecular crowding plays a crucial role in F-actin bundling. For example, in the reconstitution assays in Figure 1, the authors use 10 µM of each component of PSD (total of 60 µM), to which 5 µM actin is added. Yet, in their control assays (Supp. Figure 1), only 10 µM of each protein was checked with the same amount of actin. A control is missing where the total protein crowding would be preserved, for example, by adding BSA or protein to mimic non-specific protein crowding.

4) Is the F-bunding observed under these physiological ratios of PSD proteins and actin? For instance, a recent quantitative study (PMID: 34168338) suggests actin:Homer-1 is 200:1 or 100:1, which is in stark difference from the 1:2 molar ratio used in the study. The protein concentrations (molar ratios) need to match the physiological.

5) The imaging assay in hippocampal neurons uses an increased spine head size as a proxy for F-actin bundling. However, one needs to be careful as the baseline includes soluble mCherry, which is both much smaller in size and does not specifically enrich in the spines. The image of Homer 1 R3E shows overall lower localization at the spines. Thus, one cannot exclude that the spine enlargement upon overexpression of Homer 1 wt and R3E+EN is not primarily driven by their overall enrichment in the PSD phase. A suitable control for this assay would be mCherry-tagged PSD95, which would localize to the spines yet is not directly involved in F-actin bundling.

*Reviewer #1 (Recommendations for the authors):*

Specific Recommendations:

1. Quantitative assessment of in vitro experiments, statistical analyses: several of the in vitro experiments might gain from statistical analyses; examples are Figures 3E, 3G, 4B, 4I, and 4K. These experiments are used to establish effects, it is unclear why they are not statistically analyzed for the presence of differences. If the authors think that it is inappropriate to do so, it should be stated why and a word of caution on the conclusions may be appropriate.

2. Localization and roles of Homer proteins in neurons: the analyses of the experiments in Figure 6 seem preliminary. It is shown that transfected Homer1 (WT, R3E, R3E+EN) induces changes in spine width monitored by co-transfected GFP. R3E disrupts the ability of Homer1 to enhance spine head width, and restoring actin binding via the addition of EN restores it partially. It is important to assess whether the three Homer versions used in this experiment are expressed at similar levels and localized similarly. From the presented data, it seems that these analyses should be possible without new experiments. From the sample images, it seems that the R3E mutant Homer1 does not localize to spines, while the other versions do. The interpretation of this experiment is very different if actin-binding of Homer1 is necessary for the delivery of Homer1 to spines vs. Homer1 is localized to spines independent of its ability to bind to actin, but spine head width is controlled by Homer's ability to bind to actin.

---

## [Author Response]

Essential revisions:There are several issues that can be further improved by additional control experiments or discussions on the shortcomings of the current form (if you do not want to provide further experimental evidence).

We are extremely grateful to the encouraging comments and insightful suggestions from the reviewers and the editor. During the revision, we have addressed all of these comments and requests raised by new experiments, new analyses, and text revisions as detailed below.

1) Quantitative assessment of in vitro experiments, statistical analyses: several of the in vitro experiments might gain from statistical analyses; examples are Figures 3E, 3G, 4B, 4I, and 4K. These experiments are used to establish effects, it is unclear why they are not statistically analyzed for the presence of differences. If the authors think that it is inappropriate to do so, it should be stated why and a word of caution on the conclusions may be appropriate.

We thank the reviewers for raising this important issue. We performed statistical analysis of all the results as the reviewers asked. The details are included in the updated figures. The updated figures include Figures 3C 3E, 3G, 4B, 4I, and 4K (with corresponding descriptions added in the text of the manuscript). Student t-tests was used for all the analyses. These statistical results support the conclusions that we have drawn in the original version of the manuscript.

2) Localization and roles of Homer proteins in neurons: the analyses of the experiments in Figure 6 seem preliminary. It is shown that transfected Homer1 (WT, R3E, R3E+EN) induces changes in spine width monitored by co-transfected GFP. R3E disrupts the ability of Homer1 to enhance spine head width, and restoring actin binding via the addition of EN restores it partially. It is important to assess whether the three Homer versions used in this experiment are expressed at similar levels and localized similarly. From the presented data, it seems that these analyses should be possible without new experiments. From the sample images, it seems that the R3E mutant Homer1 does not localize to spines, while the other versions do. The interpretation of this experiment is very different if actin-binding of Homer1 is necessary for the delivery of Homer1 to spines vs. Homer1 is localized to spines independent of its ability to bind to actin, but spine head width is controlled by Homer's ability to bind to actin.

We thank the reviewer for raising this insightful point. It is important to know the impact of the mutations on Homer1’s synaptic targeting before assessing their potential roles in synaptic functions. We quantified the spine-to-shaft signal ratio of each construct expressed in the cultured neurons. Consistent with the observation of the reviewer, we found that Homer1-WT and Homer1-R3E+EN are more enriched in synapses than mCherry or Homer1-R3E (updated Figure 6). The result suggests that disrupting Homer/actin interaction can weaken the synaptic localization of Homer in spines, and thus the Homer1-R3E mutant is less capable in promoting spine head enlargement.

3) It is well established that molecular crowding plays a crucial role in F-actin bundling. For example, in the reconstitution assays in Figure 1, the authors use 10 µM of each component of PSD (total of 60 µM), to which 5 µM actin is added. Yet, in their control assays (Supp. Figure 1), only 10 µM of each protein was checked with the same amount of actin. A control is missing where the total protein crowding would be preserved, for example, by adding BSA or protein to mimic non-specific protein crowding.

We appreciate the reviewer for raising this point. We have performed a control experiment during the revision as suggested by the reviewer. To check whether higher concentrations of total proteins may also facilitate F-actin bundling, we performed a control experiment in which we mixed 5 µM actin with 60 µM Homer3 or 60 µM BSA. We did not observe any F-actin bundling under these crowded conditions (added as Figure S1B in the revised manuscript), further supporting that phase separation of Homer with other PSD scaffold proteins are essential for Homer-mediated actin bundling.

4) Is the F-bunding observed under these physiological ratios of PSD proteins and actin? For instance, a recent quantitative study (PMID: 34168338) suggests actin:Homer-1 is 200:1 or 100:1, which is in stark difference from the 1:2 molar ratio used in the study. The protein concentrations (molar ratios) need to match the physiological.

Indeed, actin is a super-abundant synaptic protein. However, most of actin would be in polymerized filamentous forms, and monomeric actin concentration in spines should not be very high. Our in vitro actin bundling assay is to test whether the Homer1-containg PSD condensate can facilitate actin polymerization and F-actin bundling. Thus, we only used a low actin monomer concentration (5 µM) for our assay. If the Homer1-containg PSD condensate can bundle actin filaments at this low actin concentration, similar actin bundling should occur when the free actin concentration is higher than 5 µM. Nonetheless, we agree with the reviewer that the ratio of PSD proteins to actin monomer in our assay may affect actin polymerization. We looked into this issue during the revision. A protein ratio of 100:1 or 200:1 (actin:PSD proteins) is technically impractical in our assay system. Instead, we tried a 1:0.5 molar ratio (i.e., 5 µM actin with 2.5 µM Homer/Shank/GKAP complex). The PSD condensates mediated F-actin bundling was robustly observed (see Figure 1—figure supplement 1C).

5) The imaging assay in hippocampal neurons uses an increased spine head size as a proxy for F-actin bundling. However, one needs to be careful as the baseline includes soluble mCherry, which is both much smaller in size and does not specifically enrich in the spines. The image of Homer 1 R3E shows overall lower localization at the spines. Thus, one cannot exclude that the spine enlargement upon overexpression of Homer 1 wt and R3E+EN is not primarily driven by their overall enrichment in the PSD phase. A suitable control for this assay would be mCherry-tagged PSD95, which would localize to the spines yet is not directly involved in F-actin bundling.

We did find that Homer1-WT and Homer1-R3E+EN are more enriched in the synapse than mCherry or Homer1-R3E (see updated Figure 6 and our corresponding description in point #2 above). This result suggests that actin-binding is required for Homer to localize in synapses. The dendritic spines are sensitive to PSD protein concentrations and overexpression of any of the scaffold proteins (PSD-95, GKAP, Shank, Homer) used in this study would increase the spine size. Thus, these proteins might not be good controls for the assay in this study. One way to mitigate this is to quantify the spine targeting of each Homer1 mutant as described in our response to the point #2 above.